# Identification of mental health and quality of life outcomes in primary care databases in the UK: a systematic review

Helena Carreira,[1] Rachael Williams,[2] Helen Strongman,[1] Krishnan Bhaskaran[1]

¹Department of Non-communicable Disease Epidemiology, London School of Hygiene and Tropical Medicine Faculty of Epidemiology and Population Health, London, UK
²Clinical Practice Research Datalink, Medicines and Healthcare Products Regulatory Agency, London, UK

**Correspondence to**
Helena Carreira;
helena.carreira@lshtm.ac.uk

## ABSTRACT

**Objectives** To summarise the definitions and combinations of codes used to identify outcomes of anxiety, depression, fatigue, cognitive dysfunction (including mild cognitive dysfunction and dementia), sexual dysfunction, pain, sleep disorders, and fatal and non-fatal self-harm in studies using electronic health records from primary care databases in the UK.

**Design** Systematic review.

**Data sources** Medline, Embase and lists of publications of the main primary care databases in the UK.

**Eligibility criteria** Included data from a UK primary care database and studied outcome(s) of interest.

**Data extraction and synthesis** We abstracted information on the outcomes definition and codelists. When necessary, authors were contacted to request codelists.

**Results** 120 studies were eligible. Codelists were available for 17/42 studies of depression; 21/41 studies of fatal and non-fatal self-harm; 17/27 studies of dementia/cognitive dysfunction; 5/12 studies of anxiety; 4/8 studies of pain; 3/6 studies of fatigue and sexual dysfunction; 1/2 studies of sleep disorders. Depression was most often defined using codes for diagnoses (37/42 studies) and/or antidepressants prescriptions (21/42 studies); six studies reported including symptoms in their definition. Anxiety was defined with codes for diagnoses (12/12 studies); four studies also reported including symptoms. Fatal self-harm was ascertained in primary care data linked to the Office for National Statistics mortality database in nine studies. Most studies of cognitive dysfunction included Alzheimer's disease, and vascular and frontotemporal dementia. Fatigue definitions varied little, including chronic fatigue syndrome, neurasthenia and postviral fatigue syndrome. All studies of sexual dysfunction focused on male conditions, principally erectile dysfunction. Sleep disorders included insomnia and hypersomnia. There was substantial variability in the codelists; validation was carried out i21/120 studies.

**Conclusions** There is a need for standardised definitions and validated list of codes to assess mental health and quality of life outcomes in primary care databases in the UK.

## INTRODUCTION

Primary care databases of electronic health records (EHRs) in the UK such as The Clinical Practice Research Datalink (CPRD), QResearch or The Health Improvement

### Strengths and limitations of this study

► Comprehensive systematic review of the literature aiming at describing the definitions and combination of codes used to identify outcomes of mental health and quality of life in electronic health records databases in the UK.

► Potential for error in the selection of the eligible studies minimised by duplication of the screening.

► The authors of the original studies were contacted to obtain the list of Read codes used when these were not publicly available.

► We only considered definitions of study outcomes, and did not consider studies where mental health or quality of life variables were covariates or exposure variables, limiting the generalisability of our results to these other contexts.

Network (THIN) have been widely used to study mental health outcomes such as depression,[1 2] and other key aspects of quality of life (QoL), such as fatigue and pain,[3 4] even though the identification of patients with these conditions is not straightforward.

Strategies to identify patients with a given condition in the EHRs typically include generating lists of relevant codes, then searching the patients' record for these codes to identify symptoms, diagnoses, referrals, appointments for disease management and monitoring, and/or prescriptions of interest.[5] The process of developing a list of codes of interest, and deciding how to apply them, may be subjective. For example, a study on the selection of codes for stroke, a relatively well-defined clinical outcome, showed that researchers with clinical and epidemiological experience may have differing interpretations of the relevance of each code.[6] A systematic review on the identification of patients with cancer in UK primary care databases described several combinations of Read codes used across studies.[7] Estimates of validity of diagnoses in these databases have been generally high across disease types,[8 9] but the heterogeneity in the codelists raises issues of misclassification, and hampers

the comparability of studies using the same data to assess the same outcome.[10] The pattern of use of the codes by the general practitioners (GPs) also needs consideration. For example, in recording depression, it has been shown that GPs have switched from diagnostic to symptom codes in recent years[11]; this may have a large impact on outcome definitions based around diagnostic codes. In addition, outcome definitions using prescription data may lead to misclassification where drugs have multiple indications: for example, sertraline, paroxetine or escitalopram, among the most commonly used antidepressants, are also first-line treatments for generalised anxiety disorder[12]; and amitriptyline, a tricyclic antidepressant, is also a first-line treatment for neuropathic pain.[13]

Given the broad interest in mental health and QoL outcomes, and the strong potential for primary care data to contribute to studying these outcomes, our aim was to systematically review and summarise the strategies used to define such outcomes in previous studies, and the extent to which case definitions have been validated.

## METHODS

This review followed the a priori defined methods specified in the systematic review protocol (online supplementary appendix 1).

### Outcomes of interest

The outcomes of interest for this review were: anxiety, depression, fatigue, cognitive dysfunction, pain, sexual dysfunction, sleep disorder and fatal and non-fatal self-harm. We considered that a study provided data for cognitive dysfunction when dementia, mild cognitive impairment or single domains of cognitive function were studied (ie, attention, executive function, memory, language, motor and social). Composite outcomes of two or more of these outcomes (eg, psychological impairment defined by anxiety or depression) were also eligible.

### Information sources and search strategy

We searched MEDLINE and Embase via Ovid, from inception up to 28 June 2018, to identify studies that involved EHRs from primary care databases and studied one of the outcomes of interest (see above). The search expressions are provided in online supplementary appendix 1, and combined terms to identify primary care databases, terms to identify mental health and QoL outcomes, and terms indicating UK-based research. The CPRD, THIN and QResearch list of publications, available in their websites, were manually revised to identify additional studies. The lists of bibliographical references of the studies considered eligible for the review were also screened by hand to identify additional studies.

### Studies eligibility

We considered eligible the studies that used data from a primary care database that routinely gathers EHR data from primary care practices in the UK, and in which the outcome of interest was one of those of interest for this study (see list above). This included purely descriptive studies on the incidence/prevalence of the outcome and analytical studies where the condition of interest was one of the main outcomes of the study. Studies of primary care data linked to other sources of data, such as the Hospital Episode Statistics (HES) or the Office for National Statistics (ONS) mortality data, were also considered eligible.

Abstracts from conferences were excluded, as it was unlikely that the methods section would provide sufficiently detailed information on the definition of the outcomes. Studies of pain caused by infectious agents (eg, herpes zoster) were excluded; similarly, studies of sleep apnoea and narcolepsy were excluded due to their unlikely psychological origin.[14 15] Studies reporting only on patterns of treatment of the conditions of interest were excluded, unless pharmacological treatment was clearly used as a proxy for the definition of the condition. Studies, where the outcome of interest was comorbidity, were also excluded. Where there were multiple studies from the same group of authors, we considered these separately, since the definition of the same outcomes could have been updated over time.

The eligibility of the studies was determined by two authors (HC and HS) reviewing all records retrieved from the publications databases. First, the title of each study was read to determine the eligibility for the review; when the information provided in the title was insufficient for a clear exclusion of the study, the study was considered for further assessment. Second, the full text of each study not previously excluded was read, in order to determine the eligibility. Disagreements over study eligibility between the two reviewers were resolved by discussion, including with a third researcher (KB or RW) where needed.

### Data acquisition and extraction

We abstracted data on study characteristics (title, study design), the primary care database used, any database(s) linked to the primary care data, outcome(s) reported, definition of the outcome(s) (ie, Read codes, drug prescriptions, International Classification of Diseases (ICD) codes, etc) and any codelist available. When there were two or more definitions of the outcome (eg, used in sensitivity analyses), we abstracted all information but considered only the main outcome for data analysis in this review. We also abstracted data on whether the codelist had been validated, and any description related to the handling of past or prevalent (at baseline in cohort studies) episodes of these outcomes. We considered that the study had attempted to validate the list of codes when the results were compared with data from another source, or when outcomes were confirmed by enquiring the patients' GP or by reviewing the patients' medical record. The data extraction process was repeated by a second author (HS) for 10% of the papers included for each outcome, to check for reliability in the extraction process.

When a study did not provide the codelist for the definition of the outcome in the original publication or in a publicly available repository, we contacted the

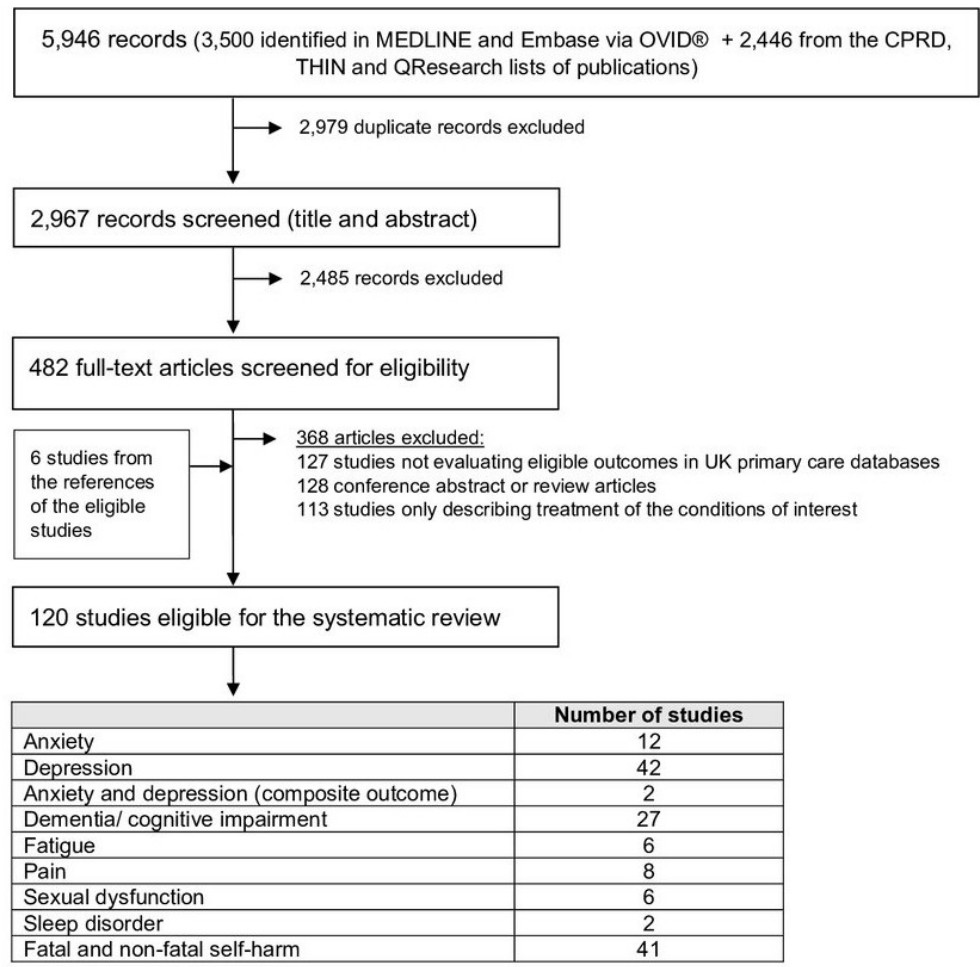

Figure 1 Systematic review flow chart. CPRD, Clinical Practice Research Datalink; THIN, The Health Improvement Network.

corresponding author of the study by email seeking this information (online supplementary appendix 2). In the case of emails that could not be delivered to the corresponding author, we searched the contact of another study author, usually the first or the last author, and addressed the email to her/him; if this failed to be delivered, no further attempt of contact was made. For all delivered emails, if no response was received within 2 weeks, a follow-up email was sent.

### Data analysis
We produced descriptive tables showing the number and proportion of studies eligible for each outcome, by primary care database and codelist availability. We described, for each outcome, the types of codes used in the definition of the outcomes (eg, diagnosis codes, symptom codes, prescription codes). The lists of codes were also reviewed to assess the clinical characteristics of the disorders included (eg, whether mixed anxiety and depression was included in the definition of anxiety or depression); this was done by manually reviewing the list of codes to identify codes related to different clinical characteristics of the specific outcome. To describe the Read codes most commonly used

to identify these outcomes in the data, we produced a list of Read codes sorted by number of studies that used the code. The results of the validation studies described in the original papers were reported descriptively.

### Patient and public involvement
No patients or public were involved in the design and conduct of this study.

### RESULTS
Of 5946 records initially identified in the bibliographical references search, 2979 were discarded as being duplicated, which left 2967 records to be assessed for eligibility (figure 1). The title assessment resulted in the exclusion of 2485 records, and 482 studies were considered for full-text assessment. Of these, 368 studies were excluded, mostly because they were abstracts from conferences or did not evaluate relevant outcomes. Six papers were identified from the screening of the references. A total of 120 studies were eligible for the systematic review; a list of codes were obtained for nearly half of the studies. The definitions and combinations of codes used to identify

**Table 1** Characteristics of the studies included in the systematic review

| | Anxiety | | Depression | | Anxiety and depression (composite outcome) | | Dementia/ cognitive impairment | | Fatigue | | Pain | | Sexual dysfunction (male) | | Sleep disorders | | Fatal and non-fatal self-harm | |
|---|---|---|---|---|---|---|---|---|---|---|---|---|---|---|---|---|---|---|
| | N | % | N | % | N | % | N | % | N | % | N | % | N | % | N | % | N | % |
| Total no of studies | 12 | 100.0 | 42 | 100.0 | 2 | 100.0 | 27 | 100.0 | 6 | 100.0 | 8 | 100.0 | 6 | 100.0 | 2 | 100.0 | 41 | 100.0 |
| **Main primary care database** | | | | | | | | | | | | | | | | | | |
| CPRD | 7 | 58.3 | 24 | 57.1 | 0 | 0.0 | 21 | 77.8 | 5 | 83.3 | 5 | 62.5 | 4 | 66.7 | 1 | 50.0 | 31 | 75.6 |
| PCCIU | 0 | 0.0 | 1 | 2.4 | 0 | 0.0 | 0 | 0.0 | 0 | 0.0 | 0 | 0.0 | 0 | 0.0 | 0 | 0.0 | 0 | 0.0 |
| THIN | 5 | 41.7 | 10 | 23.8 | 1 | 50.0 | 5 | 18.5 | 1 | 16.7 | 0 | 0.0 | 2 | 33.3 | 0 | 0.0 | 6 | 14.6 |
| Other | 0 | 0.0 | 7 | 16.7 | 1 | 50.0 | 1 | 3.7 | 0 | 0.0 | 3 | 37.5 | 0 | 0.0 | 1 | 50.0 | 4 | 9.8 |
| **Outcome ascertained using linked data (HES or ONS)** | | | | | | | | | | | | | | | | | | |
| Yes | 1 | 8.3 | 2 | 4.8 | 0 | 0.0 | 2 | 7.4 | 0 | 0.0 | 0 | 0.0 | 0 | 0.0 | 0 | 0.0 | 9 | 22.0 |
| No | 11 | 91.7 | 40 | 95.2 | 2 | 100.0 | 25 | 92.6 | 6 | 100.0 | 8 | 100.0 | 6 | 100.0 | 2 | 100.0 | 32 | 78.0 |
| **Code list availability** | | | | | | | | | | | | | | | | | | |
| Total no of studies with code lists available | 5 | 41.7 | 17 | 40.5 | 2 | 100.0 | 17 | 63.0 | 3 | 50.0 | 4 | 50.0 | 3 | 50.0 | 1 | 50.0 | 21 | 51.2 |
| Stated in publication | 2 | 16.7 | 7 | 16.7 | 1 | 50.0 | 10 | 37.0 | 1 | 16.7 | 2 | 25.0 | 2 | 33.3 | 1 | 50.0 | 17 | 41.5 |
| Cited another paper or web repository | 0 | 0.0 | 0 | 0.0 | 0 | 0.0 | 0 | 0.0 | 0 | 0.0 | 1 | 12.5 | 0 | 0.0 | 0 | 0.0 | 0 | 0.0 |
| Obtained from authors | 3 | 25.0 | 10 | 23.8 | 1 | 50.0 | 7 | 25.9 | 2 | 33.3 | 1 | 12.5 | 1 | 16.7 | 0 | 0.0 | 4 | 9.8 |
| Total no of studies with code lists not available | 7 | 58.3 | 25 | 59.5 | 0 | 0.0 | 10 | 37.0 | 3 | 50.0 | 4 | 50.0 | 3 | 50.0 | 1 | 50.0 | 20 | 48.8 |
| Cited another paper or web repository but couldn't obtain | 1 | 8.3 | 1 | 2.4 | 0 | 0.0 | 1 | 3.7 | 1 | 16.7 | 0 | 0.0 | 0 | 0.0 | 0 | 0.0 | 3 | 7.3 |
| Stated available on request but did not provide | 2 | 16.7 | 4 | 9.5 | 0 | 0.0 | 3 | 11.1 | 2 | 33.3 | 0 | 0.0 | 1 | 16.7 | 1 | 50.0 | 0 | 0.0 |

Continued

**Table 1** Continued

| | Anxiety | | Depression | | Anxiety and depression (composite outcome) | | Dementia/cognitive impairment | | Fatigue | | Pain | | Sexual dysfunction (male) | | Sleep disorders | | Fatal and non-fatal self-harm | |
|---|---|---|---|---|---|---|---|---|---|---|---|---|---|---|---|---|---|---|
| | N | % | N | % | N | % | N | % | N | % | N | % | N | % | N | % | N | % |
| Not mentioned in the paper and did not provide | 4 | 33.3 | 19 | 45.2 | 0 | 0.0 | 6 | 22.2 | 0 | 0.0 | 4 | 50.0 | 2 | 33.3 | 0 | 0.0 | 17 | 41.5 |
| Type of codes available* | | | | | | | | | | | | | | | | | | |
| Read/medical codes | 5 | 100.0 | 14 | 82.4 | 2 | 100.0 | 17 | 100.0 | 3 | 100.0 | 4 | 100.0 | 3 | 100.0 | 1 | 100.0 | 8 | 42.1 |
| OXMIS codes | 0 | 0.0 | 3 | 17.6 | 0 | 0.0 | 0 | 0.0 | 0 | 0.0 | 2 | 50.0 | 0 | 0.0 | 0 | 0.0 | 8 | 42.1 |
| ICD codes | 1 | 20.0 | 2 | 4.8 | 0 | 0.0 | 2 | 11.7 | 0 | 0.0 | 0 | 0.0 | 0 | 0.0 | 0 | 0.0 | 9 | 47.4 |
| Definition of the outcome† | | | | | | | | | | | | | | | | | | |
| Study described that codes for diagnosis were included | | | | | | | | | | | | | | | | | | |
| Yes [in isolation‡] | 12 [5] | 100.0 | 37 [14] | 88.1 | 2 [0] | 100.0 | 27 [19] | 100.0 | 6 [1] | 100.0 | 8 [6] | 100.0 | 5 [2] | 83.3 | 2 [2] | 100.0 | – [–] | – |
| No | 0 | 0.0 | 5 | 11.9 | 0 | 0.0 | 0 | 0.0 | 0 | 0.0 | 0 | 0.0 | 1 | 16.7 | 0 | 0.0 | – | – |
| Unclear | 0 | 0.0 | 0 | 0.0 | 0 | 0.0 | 0 | 0.0 | 0 | 0.0 | 0 | 0.0 | 0 | 0.0 | 0 | 0.0 | – | – |
| Study described that codes for symptoms were included | | | | | | | | | | | | | | | | | | |
| Yes [in isolation‡] | 4 (0) | 33.3 | 6 [0] | 14.3 | 2 [0] | 100.0 | 6 [0] | 22.2 | 5 [0] | 83.3 | 2 [0] | 25.0 | 0 [0] | 0.0 | 0 [0] | 0.0 | – [–] | – |
| No | 6 | 50.0 | 33 | 78.6 | 0 | 0.0 | 20 | 74.1 | 1 | 16.7 | 6 | 75.0 | 4 | 66.7 | 2 | 100.0 | – | – |
| Unclear | 2 | 16.7 | 3 | 7.1 | 0 | 0.0 | 1 | 3.7 | 0 | 0.0 | 0 | 0.0 | 2 | 33.3 | 0 | 0.0 | – | – |
| Study described that prescriptions were included | | | | | | | | | | | | | | | | | | |
| Yes [in isolation‡] | 1 [0] | 8.3 | 21 [6] | 50.0 | 2 [0] | 100.0 | 4 [0] | 14.8 | 0 [0] | 0.0 | 2 [0] | 25.0 | 3 [1] | 50.0 | 0 [0] | 0.0 | – [–] | – |
| No | 11 | 91.7 | 21 | 50.0 | 0 | 0.0 | 23 | 85.2 | 6 | 100.0 | 6 | 75.0 | 3 | 50.0 | 2 | 100.0 | – | – |
| Unclear | 0 | 0.0 | 0 | 0.0 | 0 | 0.0 | 0 | 0.0 | 0 | 0.0 | 0 | 0.0 | 0 | 0.0 | 0 | 0.0 | – | – |
| Handling of outcomes occurring prior to study period | | | | | | | | | | | | | | | | | | |
| Excluded | 5 | 41.7 | 15 | 35.7 | 0 | 0.0 | 21 | 77.8 | 3 | 50.0 | 3 | 37.5 | 1 | 16.7 | 1 | 50.0 | 7 | 17.1 |
| Adjusted/matched for | 2 | 16.7 | 8 | 19.0 | 0 | 0.0 | 1 | 3.7 | 1 | 16.7 | 0 | 0.0 | 3 | 50.0 | 0 | 0.0 | 10 | 24.4 |
| Stratified results | 1 | 8.3 | 3 | 7.1 | 2 | 100.0 | 0 | 0.0 | 1 | 16.7 | 0 | 0.0 | 3 | 50.0 | 0 | 0.0 | 4 | 9.8 |
| Not stated | 3 | 25.0 | 11 | 26.2 | 0 | 0.0 | 1 | 3.7 | 0 | 0.0 | 3 | 37.5 | 0 | 0.0 | 1 | 50.0 | 15 | 36.6 |
| Not applicable § | 2 | 16.7 | 8 | 19.0 | 0 | 0.0 | 4 | 14.8 | 2 | 33.3 | 2 | 25.0 | 0 | 0.0 | 0 | 0.0 | 9 | 22.0 |

Continued

**Table 1** Continued

Validation of list of codes

| | Anxiety | | Depression | | Anxiety and depression (composite outcome) | | Dementia/ cognitive impairment | | Fatigue | | Pain | | Sexual dysfunction (male) | | Sleep disorders | | Fatal and non-fatal self-harm | |
|---|---|---|---|---|---|---|---|---|---|---|---|---|---|---|---|---|---|---|
| | N | % | N | % | N | % | N | % | N | % | N | % | N | % | N | % | N | % |
| Yes | 2 | 16.7 | 5 | 11.9 | 1 | 50.0 | 5 | 18.5 | 0 | 0.0 | 3 | 37.5 | 0 | 0.0 | 0 | 0.0 | 9 | 22.0 |
| None stated | 10 | 83.3 | 37 | 88.1 | 1 | 50.0 | 22 | 81.5 | 6 | 100.0 | 5 | 62.5 | 6 | 100.0 | 2 | 100.0 | 32 | 78.0 |
| Study aiming at validating the outcome | 0 | 0.0 | 0 | 0.0 | 1 | 50.0 | 0 | 0.0 | 0 | 0.0 | 0 | 0.0 | 0 | 0.0 | 0 | 0.0 | 3 | 7.3 |

*Refers only to studies providing codelists. A study may include more than one type of codes.

†Defined based on the description of the outcome provided in the original studies. Studies were considered to have included symptoms in the definition of the outcomes when this information was explicitly stated by the authors. Pain was considered a symptomatic outcome, regardless of how it was described in the original publications.

‡The number of studies where the definition of the outcome included only this category of codes is provided in squared brackets.

§Depends on study design and outcome. For example, it includes studies describing the epidemiology of first ever diagnoses or focusing on completed suicide only.

CPRD, Clinical Practice Research Datalink; HES, Hospital Episodes Statistics; ICD, International Classification of Diseases; ONS, Office for National Statistics; OXMIS, Oxford Medical Information System; PCCIU, Primary Care Clinical Informatics Unit Research Data Set; THIN, The Health Improvement Network.

mental health and QoL outcomes from UK primary care databases were heterogeneous for all outcomes; there was particular variability in the inclusion/exclusion of codes for symptoms of the mental disorders. Prescriptions were not frequently used as proxy for mental disorders. Validation efforts were rarely employed. Detailed results for each outcome are provided below.

### Anxiety

Twelve studies had anxiety as an outcome (table 1 and online supplementary appendix 3 table 1); of these, two studied panic only.[2 16] The list of codes used to identify outcomes of anxiety was available for 5 of the 12 studies (41.7%); in one study, the cases of anxiety were identified in CPRD data linked to HES. All 12 studies included codes for diagnosis of anxiety, and 4 (33.3%) included also codes for anxiety symptoms. Prescriptions were considered in the definition of the outcome in one study only (8.3%).

Of the five studies for which codelists were available, five included codes for generalised anxiety disorder (100%), four for phobia (80.0%), four for panic disorder/attacks (80.0%), three for mixed anxiety and depression (60.0%), and two for stress-related disorders (40.0%) (table 2). Codes for post-traumatic stress disorder and obsession-compulsion were less often included (one study each, 20.0%).

Only one study reported including drugs prescriptions in the definition of the outcome[17]; this considered diazepam and lorazepam only (table 3).

Two studies[2 18] assessed the validity of the codelists (table 4). The proportion of cases confirmed was reported in one study: 73.5% for cases treated with anxiolytics, antidepressants and hypnotics, and 89.6% in those not pharmacologically treated.[18]

Online supplementary appendix 4 table 1 provides the list of Read codes used to identify patients with anxiety in the eligible studies; online supplementary appendix 4 table 2 provides the list of ICD-10 codes.

### Depression

Forty-two studies identified outcomes of depression (table 1 and online supplementary appendix 3 table 2). The list of codes used to identify outcomes of depression was available for 17 of the 42 studies (40.5%); 2 studies identified cases of depression in primary care data linked to HES data, using ICD-10 codes. Six studies defined depression by proxy of antidepressants intake only; the remaining 36 studies described to have included codes for diagnoses of depression and 6 (14.3%) studies also considered symptoms of depression in the definition of the outcome. Fifteen studies (35.7%) reported having excluded patients with history of depression.

Of the 17 studies for which the codelists were available, 10 included codes for mixed anxiety and depression (58.8%), 4 for bipolar disorder (23.5%) and 3 for depression in dementia (17.6%) (table 2).

Antidepressant prescriptions, in isolation or combination with diagnostic/symptoms Read codes, were considered in the identification of patients with depression in 21 studies (50.0%); in six studies, depression was solely

| Table 2 | Clinical characteristics of the outcomes of interest in the studies for which the list of codes was available |
| --- | --- |

| Study included codes for | N | % of total with code lists available |
| --- | --- | --- |
| Anxiety | 5 | 100.0 |
| Generalised anxiety disorder | 5 | 100.0 |
| Panic disorder/attacks | 4 | 80.0 |
| Phobia | 4 | 80.0 |
| Mixed anxiety and depression | 3 | 60.0 |
| Stress-related disorders | 2 | 40.0 |
| Obsession-compulsion | 1 | 20.0 |
| Post-traumatic stress disorder | 1 | 20.0 |
| Depression | 17 | 100.0 |
| Unipolar depression | 17 | 100.0 |
| Depression with psychotic symptoms | 14 | 82.4 |
| Mixed anxiety and depression | 10 | 58.8 |
| Bipolar disorder | 4 | 23.5 |
| Depression in dementia | 3 | 17.6 |
| Dementia/cognitive impairment | 17 | 100.0 |
| Alzheimer's disease | 13 | 81.3 |
| Vascular dementia | 13 | 81.3 |
| Frontotemporal dementia | 12 | 75.0 |
| Lewy bodies disease | 11 | 68.8 |
| Mild cognitive impairment only | 3 | 17.6 |
| Fatigue | 3 | 100.0 |
| Chronic fatigue syndrome/myalgic encephalitis | 3 | 100.0 |
| Neurasthenia | 3 | 100.0 |
| Post viral fatigue syndrome | 3 | 100.0 |
| Fibromyalgia | 2 | 66.7 |
| Pain | 4 | 100.0 |
| Chest pain | 1 | 25.0 |
| Chronic widespread pain | 1 | 25.0 |
| Musculoskeletal pain | 1 | 25.0 |
| Unspecified abdominal pain | 1 | 25.0 |
| Sexual dysfunction (male) | 3 | 100.0 |
| Erectile dysfunction | 3 | 100.0 |
| Other male sexual dysfunctions | 1 | 33.3 |
| Sleep disorder | 1 | 100.0 |
| Insomnia | 1 | 100.0 |
| Hypersomnia | 1 | 100.0 |
| Fatal and non-fatal self-harm | 21 | 100.0 |
| Completed suicide | 17 | 80.9 |
| Completed suicide only | 8 | 38.1 |
| Completed and attempted suicide only | 6 | 28.6 |

Continued

| Table 2 | Continued |
| --- | --- |

| Study included codes for | N | % of total with code lists available |
| --- | --- | --- |
| Completed and attempted suicide, and self-harm | 4 | 19.1 |
| Included deaths of undetermined intent | 7 | 33.3 |
| Attempted suicide/self-harm | 4 | 19.1 |

defined by the prescription of antidepressants (table 3). The list of antidepressant categories was seldom provided; of the studies that reported this information, selective serotonin reuptake inhibitors were the group most often considered (six studies), followed by monoamine oxidase inhibitors and tricyclic and related antidepressants drugs (three studies each).

Five studies (11.9%) assessed the performance of the list of codes to identify patients with depression (table 4). The proportion of cases confirmed was reported by two studies only: 83.3%[19] and 89.6%.[20]

The list of Read codes used to identify patients with depression is provided in online supplementary appendix 4 tables 3 and 4 provides the list of ICD-10 codes.

### Composite outcome of anxiety and depression

Two studies provided data for composite outcomes of anxiety and depression (table 1 and online supplementary appendix 3 table 3). The codelist was available for the two studies. The studies reported including codes for symptoms as well as diagnosis of anxiety and depression, and included prescriptions of antidepressants and anti-anxiety drugs in the definition of the outcome.

John et al[21] compared the performance of 12 different algorithms to identify patients with anxiety and depression in the Secure Anonymised Information Linkage Databank; the positive predictive value of the Read codes for anxiety and depression diagnoses, symptoms and treatments, against the five-item Mental Health Inventory (gold standard), varied between 61% and 76%[21] (table 4). The list of Read codes used to identify patients with composite outcomes of anxiety and depression is provided in online supplementary appendix 4 table 5.

### Cognitive dysfunction (including mild cognitive dysfunction and dementia)

Twenty-seven studies reported outcomes of dementia or cognitive function (table 1 and online supplementary appendix 3 table 4). The codelists were available for 17 studies (63.0%); in two studies dementia was ascertained in primary care data linked to other sources of data. All studies included codes for diagnosis of dementia or cognitive impairment, and six studies (22.2%) reported to have included also codes for symptoms of dementia. Twenty-one studies (77.8%) referred to have excluded

**Table 3** Pharmacological categories used in the studies that used drug prescriptions to identify patients with the outcome of interest

| Outcome | No of studies | % |
|---|---|---|
| Anxiety | 12 | 100.0 |
| Studies of anxiety that used drugs prescriptions only | 0 | 0.0 |
| Studies of anxiety that used drugs prescriptions | 1 | 8.3 |
| Diazepam and lorazepam | 1 | 8.3 |
| Depression | 42 | 100.0 |
| Studies of depression that used drugs prescriptions only | 6 | 14.3 |
| Studies of depression that used drugs prescriptions | 21 | 50.0 |
| Antidepressants, categories not further specified | 15 | 35.7 |
| Antidepressants, categories further specified | 6 | 14.3 |
| Tricyclic and related antidepressant drugs | 3 | 7.1 |
| Monoamine oxidase inhibitors | 3 | 7.1 |
| Selective serotonin reuptake inhibitors | 6 | 14.3 |
| Other antidepressant drugs | 3 | 7.1 |
| Dementia | 27 | 100.0 |
| Studies of dementia that used drugs prescriptions only | 0 | 0.0 |
| Studies of dementia that used drugs prescriptions | 4 | 14.8 |
| Anticholinesterases | 4 | 14.8 |
| Dopaminergic drugs | 4 | 14.8 |
| Pain | 8 | 100.0 |
| Studies of pain that used drugs prescriptions only | 2 | 25.0 |
| Studies of pain that used drugs prescriptions | 2 | 25.0 |
| Analgesics, not otherwise specified | 2 | 25.0 |
| Antidepressants | 2 | 25.0 |
| Antiepileptics | 2 | 25.0 |
| Anaesthetics | 2 | 25.0 |
| Sexual dysfunction (male) | 6 | 100.0 |
| Studies of sexual dysfunction that used drugs prescriptions only | 1 | 16.7 |
| Studies of sexual dysfunction that used drugs prescriptions | 3 | 50.0 |
| Phosphodiesterase type-5 inhibitors | 2 | 33.3 |
| Prostaglandin analogues and prostamides | 1 | 16.7 |

patients with prior diagnoses of dementia from longitudinal analyses.

Of the 17 studies for which the codelists were available, 13 reported codes for Alzheimer's disease (81.3%), 13 for vascular dementia (81.3%), 12 for frontotemporal dementia (75.0%) and 11 for Lewy bodies disease (68.8%) (table 2). Three studies reported data for cognitive impairment without dementia (17.6%).

Four studies (14.8%) used prescriptions in the identification of patients with dementia; all four included anticholinesterases and dopaminergic agents (table 3).

Five studies[22–26] involved validation of the list of codes; the proportion of cases confirmed varied between 74% and 100% (table 4).

The list of Read codes used in the studies is provided in online supplementary appendix 4 table 6; the list of ICD-10 codes is provided in online supplementary appendix 4 table 7.

**Fatigue**

Six studies had fatigue as the outcome (table 1 and online supplementary appendix 3 table 5). All studies considered codes for diagnoses of fatigue, and five studies also described including codes for symptoms of fatigue. The list of codes use to identify patients with fatigue was provided in three studies (50.0%).

The three studies for which the codelist was available included codes for chronic fatigue syndrome, neurasthenia and postviral fatigue syndrome (table 2). Fibromyalgia was included in two studies (66.7%). None of the

**Table 4** Methods and results of the validation of the outcomes reported in the original studies

| Outcome and study authors | Validation method | # case validations completed/# case validations attempted | % of cases confirmed |
|---|---|---|---|
| **Anxiety** | | | |
| Martín-Merino et al, 2010[18] | GP questionnaire | 135/140 | Among pharmacologically treated: 73.5%; Among not pharmacologically treated: 89.6%. |
| Meier et al, 2004[2] | Record review | nr/nr | nr |
| **Depression** | | | |
| Becker, 2011 | Sensitivity analysis with different definitions | nr/nr | nr |
| Hagberg, 2016 | Record review | nr/nr | nr |
| Martín-Merino et al, 2010[18] | GP questionnaire | 135/140 | 89.6% |
| Meier et al, 2004[2] | Record review | nr/nr | nr |
| Yang et al, 2003[19] | Record review | 30/nr | 83.3% |
| **Anxiety and depression (composite outcome)** | | | |
| John, 2016 | Compared 12 EHR algorithms to results of the Mental Health Inventory , a subscale of SF-36 | 2799* | Between 61% and 76%, depending on the algorithm. |
| **Dementia/cognitive impairment** | | | |
| Imfeld et al, 2013 and Imfeld et al, 2015[22 23] | GP questionnaire | nr/120 | Alzheimer's disease: 79%; Vascular dementia: 74%. |
| Dunn et al, 2005[24] | GP asked to confirm diagnosis | 50/200 | 100% |
| Dunn et al, 2005[26] | GP questionnaire | 95/~100 | 83% |
| Strom et al, 2015[25] | GP questionnaire | 86/100 | 88.4% |
| Strom et al, 2015[25] | Review of free text | 1047/1048 | 1.5% patients excluded as not having the diagnosis; 42.4% confirmed as having definite memory loss, 36.8% possible memory loss, 3.2% undetermined and 16.0% unknown. |
| **Pain** | | | |
| Hall et al, 2013[4] | GP questionnaire | 48/54 | 56% |
| Mansfield et al, 2017[27] | EHR data linked to self-reported pain status collected by postal questionnaire | 1780* | 97% |
| Becker, 2008 | GP questionnaire | 176/200 | 86.4% |
| **Self-harm** | | | |
| Thomas et al 2013[29] | Comparison of cases of suicide and self-harm identified in CPRD with Read codes, with the cases identified in CPRD data linked to HES data, and published self-harm incidence data. | 74236* | 68.4% |
| **Suicide (attempted and completed)** | | | |
| Hagberg, 2016 | Record review | nr/nr | nr |
| Haste, 1998 | GP asked to confirm suicides | 77% of uncertain deaths/nr | 82% |
| Jick, 1995 | Record review | nr/nr | nr |
| Meier et al, 2004[2] | Record review | nr/nr | nr |

**Table 4** Continued

| Outcome and study authors | Validation method | # case validations completed/# case validations attempted | % of cases confirmed |
|---|---|---|---|
| Schuerch, 2016 | Outcomes identified in CPRD were compared with those identified in CPRD linked to HES and ONS data. | nr/nr | Compared with CPRD data, the frequency of the outcomes in linked data was approximately three times higher. |
| Yang et al, 2003[19] | Record review | 30/nr | 83.3% |
| Suicide (completed) | | | |
| Arana, 2010 | GP questionnaire and record review | nr/132 | 97% |
| Arana, 2010 | GP questionnaire and record review | nr/86 | 87% |
| Hall, 2009[28] | GP questionnaire and record review | 33/33 | 21.2% |
| Thomas et al, 2013[29] | Comparison of cases of suicide and self-harm identified in CPRD with Read codes, with the cases identified in CPRD data linked to ONS mortality data, and national suicide rates. | 1767* | 59.7% for men; 46.0% for women. |

*Validation attempted and completed for all patients identified in electronic health records database.
CPRD, Clinical Practice Research Datalink; EHR, electronic health record; GP, general practitioner; HES, Hospital Episode Statistics; nr, not reported; ONS, Office for National Statistics; SF-36, 36-item Short Form Health Survey.

studies assessed the validity of the list of the codes. The list of Read codes used in the studies of fatigue is available in online supplementary appendix 4 table 8.

### Pain

Pain was the outcome in eight studies (table 1 and online supplementary appendix 3 table 6). The list of codes was available for four of the eight studies. Of these four studies, three looked at pain by body site (ie, chest, abdominal, musculoskeletal pain), one study studied widespread body pain (table 2). Two studies included drugs in the identification of patients with pain; all considered antiepileptics (in the absence of codes for an epilepsy diagnosis), anaesthetics, antidepressants and analgesics (table 3).

Three studies validated the list of patients selected with the codelist (table 4). The proportion of cases confirmed varied between 56% and 86.4%. One study compared pain recorded in the EHR with pain reported in a survey; in 97% of the self-reported cases of pain, there was an entry in the EHR.[27]

Online supplementary appendix 4 table 9 provides the list of codes used in the original studies.

### Sexual dysfunction

Six studies had sexual dysfunction as an outcome, all of which focused on male sexual dysfunction (table 1, online supplementary appendix 3 table 7). Three studies provided codelists (50.0%). Of these, all included codes for erectile dysfunction and one study included codes for other male sexual dysfunctions (table 2). Three studies included considered the prescription of drugs sufficient to ascertain the outcome; two studies considered phosphodiesterase type-5 inhibitors (table 3). No study validated the list of codes used. The list of Read codes used in

the original studies is available in online supplementary appendix 4 table 10.

### Sleep disorders

Two studies were eligible for sleep disorders (table 1, online supplementary appendix 3 table 8); the two studies included diagnoses of insomnia, and one included hypersomnia as well (50%) (table 2). The list of codes was available for one study. No validation was reported. Online supplementary appendix 4 table 11 provides the list of Read codes used in the original study.

### Fatal and non-fatal self-harm

Forty-one studies had outcomes related to fatal and non-fatal self-harm (table 1 and online supplementary appendix 3 table 9). The list of codes used to define the outcomes was available for 21 studies (51.2%); 9 studies reported using ICD-10 codes.

Of the 21 studies for which the codelist was available, 17 studies (80.9%) included completed suicide, while 4 studies focused on attempted suicide only (19.1%). Of the 17 studies including completed suicide as an outcome, eight reported only completed suicides, six considered completed and attempted suicides and four included complete and attempted suicide, as well as self-harm (table 2). All studies where outcomes were identified using primary care data linked to ONS mortality data (gold standard) considered deaths recorded as of undetermined intent in the definition of suicide.

Nine studies involved some method of validation of the list of cases identified via code search (table 4). Four studies referred to have revised the clinical record of the patient to determine the final outcome and two studies asked the GPs to confirm the events. The proportion of

cases confirmed varied between 21.2% and 97%. Hall[28] assessed the validity of cause of death recording in the THIN primary care database through search of the free text and death certificate review; the underlying cause of death registered in the death certificate was listed as the cause of death in the EHR in 70% of the cases. Thomas *et al*[29] compared the ascertainment of cases of suicide and self-harm using Read codes in CPRD, with those ascertained when data from HES and ONS mortality data were available. 26.1% of the cases of suicide identified in the ONS mortality data were registered in the CPRD primary care database. HES was considered the gold standard for self-harm; 68.4% of the cases of self-harm in HES were identified as such in CPRD.

Online supplementary appendix 4 table 12 provides the list of Read codes used to identify outcomes of fatal and non-fatal self-harm in primary care data; online supplementary appendix 4 table 13 includes the lists of ICD-10 codes using in studies of linked data.

## DISCUSSION
### Results overview

This review summarised the definitions and combinations of codes used to identify outcomes of anxiety, depression, dementia and cognitive impairment, fatigue, pain, male sexual dysfunction, sleep disorder and self-injurious behaviour in primary care databases of patients in the UK. The list of codes used in the original studies was obtained for approximately half of the papers; the lack of detailed information on the definition of the outcomes in most studies raises important questions as to whether studies can be replicated by others. In the studies where the codelist was available, for all outcomes, there was substantial heterogeneity in the type of codes included (eg, diagnoses and symptoms) and drugs selected to identify outcomes; for the remaining studies, the details provided in the original publications suggest a similar pattern. We also noted considerable variability in the clinical definition of some outcomes (eg, inclusion/exclusion of bipolar disorders in studies of depression). Validation of codes used to identify these outcomes was rarely carried out; where done, positive predictive values of case definitions were variable but mostly above 80%. To overcome these issues in the current context of limited number of studies with validation efforts, it is imperative that researchers develop, validate and make publicly available code lists for these outcomes.

### Strengths and limitations

This review is based on an extensive search of the studies involving EHRs in the UK. Errors in study selection and data extraction were minimised by the independent assessment of the studies by two investigators. We contacted the authors of all original studies where the list of codes had not been provided in the original publication to seek this information; this largely increased the number of studies for which lists of codes were available, and contributed to

a more detailed characterisation of the combination of codes used to define mental health outcomes in primary care databases of EHR in the UK.

However, this review has limitations. Some relevant studies may have been missed due to imperfect search terms, as there is no Medical Subject Headings (MeSH) term for the primary care databases, and studies could be potentially missed if the keywords did not appear in the title and abstract, or due to inaccurate indexing in the publications database. We attempted to minimise the risk of missing potential eligible studies by using broad search terms incorporating both indexing terms and keywords, two databases with different indexing systems, and an additional manual check of the eligible studies and list of bibliographical references from the main EHR databases. We only considered studies where mental health or QoL variables were the outcomes of interest, limiting generalisability to other contexts. For example, we excluded studies where these variables were covariates because we expected that detailed information about covariate definitions would rarely be available. We also excluded studies where the mental health or QoL variable was used to define the patient population (eg, a study of risk of stroke in depressed patients), on the basis that decisions about how to define cases may have had quite different motivations, compared with studies where the condition was the outcome of the study, making case definitions difficult to meaningfully compare. We included studies that explicitly referred to using prescription data as a proxy for the definition of the condition (eg, treated depression assessed by proxy of antidepressant intake), but we acknowledge that it was not always clear to decide whether treatment of the condition was being used to define the condition. This could have resulted in a few studies erroneously excluded, even though this should have been minimised by the duplication of the search and study selection process by two researchers working independently, with discussion of all discordant results. It is unclear if the list of codes that could not be obtained differ in any systematic way to the ones obtained. Some authors expressed concerns over intellectual property when sharing the list of codes, and this may have been a bigger concern among those who put a lot of time and thought into their codelists; on the other hand, authors who have concerns about the quality of their codelist may have been less willing to share them. Lastly, we summarised the types of codes used to define the outcome based on what was stated in original studies' methods sections (because code lists were not available for all studies), but this may have been inaccurate, for example, some studies that reported in their methods only including diagnosis codes then provided code lists that appeared to also contain symptom codes.

### Availability of the list of codes

The list of codes was provided in the original publications for just over a quarter of the studies. Contacting the authors resulted in codelists being made available for approximately half of the studies. For the remaining

studies, the authors either could not be contacted (eg, moved institutions, retired) or could not locate the relevant codelist (including for some studies where the paper had stated that the codelist would be available on request). Provision of codelists within the publication or in a web repository would eliminate the difficulties of authors having to be contacted and archived codelists retrieved. Most journals currently accept codelists in online supplementary appendices. Codelists were hardly ever obtained for older studies, especially those published before 2000, when email addresses were not routinely included in the details of the corresponding authors. We searched for alternative contacts in these cases, but not always successfully.

## Variability in the definition of cases and codelists

### Anxiety and depression

Anxiety was often defined with diagnostic and symptoms codes, and in a few studies by the prescription of anxiolytics and hypnotics. Even though the sensitivity of symptoms codes for anxiety is expected to be high, the positive predictive value is unknown. Anxiolytics may also result in misclassification of the outcomes, as they are currently discouraged as first line of treatment for anxiety[12] and are often prescribed for management of other conditions such as insomnia. No study considered antidepressants in the definition of anxiety even though these are currently used to manage anxiety[12]; this may have resulted in cases of anxiety treated with antidepressants, and where no Read code was available, being missed.

The inclusion/exclusion of codes for symptoms may have a larger impact in the definition of depression, as it has been shown that GPs switched from diagnostic to symptoms codes after the introduction of performance indicators in the GP contract Quality and Outcomes Framework in 2006[11] and under claims that depression was being overdiagnosed.[30 31] Codelists solely relying on Read codes for diagnosis of depression are, therefore, likely to have low sensitivity, but the impact of including/excluding a specific code will be variable, depending on how often that code is used by GPs at the point of providing care. In a few studies, depression was defined by proxy of antidepressant prescribing, alone or in combination with Read codes for symptoms/diagnosis. Considering antidepressant prescribing in the definition of depression has several issues. Certain types of antidepressants are currently used as first line of treatment for other conditions, such as pain and anxiety, and the studies relying solely on this information will be affected by misclassification of the outcome; some studies took this into account by excluding low dose tricyclic antidepressants, usually prescribed for pain, from their list of codes used to define depression.[32–34] Among the studies that did include antidepressants in their definition, there was heterogeneity in the group of antidepressants included, with some studies selecting only a few specific drugs commonly used for the treatment of depression. Studies defining depression by proxy of antidepressant prescribing only are likely also to be affected by changes in the behaviour of antidepressant prescribing. In 2004, the National Institute for Health and Care Excellence (NICE) issued guidelines discouraging antidepressants for mild depression,[35] and in 2006 a performance indicator in the UK GP Quality and Outcomes Framework pay for performance was introduced for depression severity assessed with validated symptoms questionnaires.[36] Following this measure, the proportion of new cases of pharmacologically treated depression decreased (from 73% in 2003 to 61% in 2012[37]), but the proportion of recurrent episodes pharmacologically treated increased from 74.3% to 77.8%.[37] Treatment duration times with antidepressants also increased over time[38]; this may affect the number of new episodes of depression identified in the studies. In several studies, the authors chose to report separate results for antidepressant prescribing, without using this information to ascertain the outcome of depression[39 40]; this may partially be due to the difficulties of ascertaining the indications for which antidepressants were prescribed. John *et al* explored the indications of antidepressants; more than half of the new antidepressant prescriptions were for depression, with increasing but low incidence of prescriptions for pain and anxiety, but the authors could not identify the indication for antidepressants in 17% of the new prescriptions.[41]

Regardless of the type of codes included, authors will need to often choose the inclusion/exclusion of codes relating to the clinical profile of the patients. This may have a particular impact for conditions that are highly comorbid. For example, the code for 'mixed anxiety and depression' was sometimes used in the definition of anxiety and in the definition of depression; anxiety and depression are highly comorbid and the inclusion/exclusion of these patients may have an impact on the results. In addition, for depression, the inclusion of codes related to depression in the context of bipolar disease, dementia and schizophrenia may raise issues as to whether it represents a primary depressive episode.

Part of the heterogeneity in the list of codes used to identify these outcomes may be explained by the complexity of these conditions and by the purpose for which these data are collected. Electronic healthcare data are primarily collected to provide patients with treatment, and distinctions between diagnosis and symptoms may have less weight at the point of care than when researchers aim to define these conditions using data routinely collected.

### Fatal and non-fatal self-harm

Routinely collected primary care data were shown to have low sensitivity to detect cases of suicide.[29] Thus, record linkage to ONS mortality data is of interest; this has the advantage of including causes of death other than suicide. Ascertainment of the cause of death is not always straightforward when the death is non-natural, and several studies have included cases of accidents and open verdicts in their case definition. Open verdicts have been shown to include many similarities with suicides, and several are

later registered as suicides; these are recommended to be included in studies of suicide.[42] Studies varied on whether cases of self-harm without suicidal ideation were included (eg, Rubino *et al* reviewed free text to exclude those who did not seem to have attempted suicide[43]). For self-harm, linkage to HES data will allow for more cases to be identified,[29] even though authors must consider the balance between reduction of sample size and ascertainment of the outcome, as linkage is only available for a subset of patients.

### Pain

The aetiology and location of pain in the studies involved in this review varied due to our broad inclusion criteria. When pharmacological treatment was included in the definition of pain, this was most often done with prescriptions of antidepressants and antiepileptics. Antidepressants such as first-generation tricyclic antidepressants have been used for over 30 years to manage neuropathic pain (eg, amitriptyline, doxepin, clomipramine and dosulepin).[44] Antiepileptic drugs reduce neuronal excitability and alleviate pain through several mechanisms.[44]

### Other outcomes

We considered cognitive dysfunction as a composite outcome including studies from mild to severe impairments such as those in dementia; between 10% and 20% of the patients with mild cognitive impairment are expected to convert to dementia.[45 46] Fewer studies had fatigue, sexual dysfunction and sleep disorders as the outcome, and no study was eligible for female sexual dysfunction. The definition of these outcomes varied little across the studies but the small numbers preclude firm conclusions. It has been reported that chronic fatigue increased prior to 2001,[3] but decreased between 2001 and 2013,[47] possibly due to the introduction of diagnostic criteria from NICE[48]; in the same period, increases were noted in the diagnoses of fibromyalgia.[47] This may reflect the complexity of diagnosing fatigue, which is done by exclusion of other causes only.[48]

### Validation

Outcomes identified in EHRs may lack of validity: a person meeting the operational definition for the outcome based on specific codes may not have the diagnosis or vice versa. Only a small number of studies assessed validity in their studies, and this was almost always about assessing positive predictive value of the case definition, with sensitivity and specificity rarely explored. Of these, some studies only stated that validation had been carried out, but did not report the results, which makes the performance of the case definition unclear. However, the studies that reported results tended to show a high proportion of cases confirmed by their primary care physician or by further investigations (ie, a high positive predictive value). This is in accordance with the results of two systematic reviews that assessed the validity of the diagnostic coding within the CPRD primary care database.[8 9] Studies in which

identified cases were validated by the GP did not usually specify how this validation was done—that is, whether the GPs confirmed cases by consulting the EHR, referring to additional information, relying on memory or using other methods. If GPs simply checked the same EHR used to identify the case in the first place, resulting estimates of positive predictive value would be expected to be high, but may be misleadingly optimistic.

### Implications

Mental health and QoL-related outcomes are difficult to identify in EHR databases; and thus, extra care needs to be used when defining these outcomes. The use of a particular code can vary between GP practices; for example, a study on the interpractice use of Read codes for diabetes showed that the most generic code was used in 14%–98% of the patients with diabetes in the practices.[49] GPs can derive Read codes for their practice; this may raise issues with new codes being added over time,[50] and codelists that need to be updated. It is important that authors clearly document the process of selection of the codes, so that these are available with clear rationale if needed.[5] Repositories of lists of codes allow researchers to access codelists easily. However, these repositories also need funding to be maintained, which limits their stability and consequently their use. Some studies of depression and dementia referred to using the Read codes recommended by the Quality and Outcomes Framework[36]; these are likely to be highly specific. It is also important to better understand the patterns of recording of some of these conditions, as changes in the patterns of use of the codes may have impact in the list of codes chosen. The inclusion of codes for symptoms and prescriptions must consider what is known about the use of codes by GPs at the point of patient providing care, as data recording in this setting is primarily intended to support clinical care. Future works are needed to understand how GPs conceptualise mental health problems, as these are expected to have less stringent definitions than psychiatrists, and this could provide insights into more meaningful case definitions.

Validation of the outcomes appears to be essential to understand the validity of case definitions. A balance between sensitivity and specificity may be considered depending on the aim of the study[5]; for depression, for example, the inclusion of terms such as 'low mood' may increase sensitivity, at the expense of decreased specificity, as some individuals who would not fit more stringent criteria for a diagnosis will be incorrectly classified as depressed.[5] A particular challenge with validation of primary care-based mental health outcomes is quantifying false negatives, which requires linkage to a high-quality external source of information, to identify cases that may have been 'missed' in primary care records. The Mental Health Dataset, which includes individual patient records of adults seeking mental health services in secondary care and has recently been made available for linkage with CPRD primary care databases, represents an opportunity

to assess the proportion of false negatives identified with the code lists, at least for more severe outcomes. Until then, sensitivity analysis using different lists of codes should be done, so that results can be compared and the impact of using different code lists evaluated. The consequences of underascertaining mental health outcomes are likely to depend on study design; in a cohort design this will not generally result in biased relative risks, whereas in a case–control context, a bias towards the null is likely. Studies might consider to use internal validation strategies, by assessing the proportion of patients referred for treatment or prescribed a relevant pharmacological agent.

Primary care databases of EHRs have made important contributions to medicine worldwide, particularly in the fields of infectious, respiratory and cardiovascular diseases. The burden of mental disorders in high-income countries has increased substantially in the last decades,[51] and more research is needed to be better understand these conditions. Primary care databases of EHRs have potential to make huge contributions to this area but, for this to happen, we need coordinated efforts across funding and research organisations to improve data quality. For example, if scientific journals make a requirement of having publicly available lists of codes, this would likely encourage researchers to spend more time defining the outcomes and potentially seek funding for validation studies, which in turn could increase the awareness of funding institutions for the importance of assessing data quality in projects using these data. In the meantime, transparency in the list of codes used to define these outcomes and reporting of sensitivity analysis with different lists of codes are key.

Despite the difficulties of assessing each separate outcome, we must take into account that mental health disorder symptoms often overlap, and is difficult to disentangle what is attributable to each condition. Lastly, these conditions have a long period of exposure to medication after symptoms have disappeared, besides a high probability of relapse and recurrence,[52] which may raise issues on whether the condition is incident or prevalent.

## CONCLUSIONS

Detailed information about codes used to identify outcomes of anxiety, depression, fatigue, cognitive dysfunction, sexual dysfunction, pain, sleep disorders, and fatal and non-fatal self-harm in studies using EHRs from primary care databases in the UK was unavailable for around half of studies of these outcomes. Where available, there was substantial heterogeneity in the list of codes used to ascertain cases. Most studies did not validate case definitions, though when this was done, positive predictive values were generally high. This review focused on common mental health disorders and QoL outcomes, but our conclusions are likely to be generalisable to other mental health outcomes. Caution is needed when interpreting and comparing results between

studies, as heterogeneity in case definitions may be large. Future studies should fully report outcomes definitions, use sensitivity analysis to mitigate uncertainties about the impact of the case definition on studies' reported outcomes, and seek to validate the list of codes used to identify these outcomes.

**Acknowledgements** We would like to thank the authors of the original studies who kindly provided the list of codes used in their studies. We would also like to thank Daniel Dedman (CPRD/LSHTM) for helpful comments on the search expressions. The copyright of the morbidity definitions/categorisations lists (©2014) obtained from the authors for the studies by Roddy *et al*, 2013, Sultan *et al*, 2017 and Mansfield *et al*, 2018 are owned by Keele University, the development of which was supported by the Primary Care Research Consortium. The authors would like to acknowledge Keele University's Prognosis and Consultation Epidemiology Research Group who have given us permission to utilise the morbidity definitions/categorisations lists (©2014); for access/details relating to the morbidity definitions/categorisation lists (©2014) please go to www.keele.ac.uk/mrr

**Contributors** HC, RW and KB designed the study. HC and HS screened the list of references and abstracted information from the original studies. HC wrote the first draft of the manuscript. All authors revised the paper for important intellectual content.

**Funding** This work was supported by the Medical Research Council (MRC) and the Clinical Practice Research Datalink (CPRD) at the Medicines and Healthcare products Regulatory Agency (MHRA) (grant number MR/M016234/1 to H.C.); and the Wellcome Trust and the Royal Society (grant number 107731/Z/15/Z to K.B.).

**Competing interests** KB reports grants from Wellcome Trust, the Royal Society, Medical Research Council and British Heart Foundation, outside the submitted work. RW reports that CPRD has financial relationships with its clients, including the London School of Hygiene and Tropical Medicine, in relation to providing access to research data and services outside the submitted work.

**Patient consent for publication** Not required.

**Provenance and peer review** Not commissioned; externally peer reviewed.

**Data sharing statement** All data relevant to the study are included in the article or uploaded as online supplementary information.

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
