## [Reviewer comments · BMJ Open]

ARTICLE DETAILS

TITLE (PROVISIONAL)	Identification of mental health and quality of life outcomes in primary care databases in the United Kingdom: a systematic review
AUTHORS	Carreira, Helena; Williams, Rachael; Strongman, Helen; Bhaskaran, Krishnan

VERSION 1 – REVIEW

REVIEWER	Elizabeth Ford Brighton and Sussex Medical School, UK
REVIEW RETURNED	06-Feb-2019

GENERAL COMMENTS	I congratulate the authors on conducting a timely and thorough piece of work on an important topic. This review of the definition of mental health outcomes in primary care databases will be a useful resource for many researchers in the UK and beyond. The authors have tackled a huge topic in a systematic way, giving insight into the heterogeneity of definitions of mental health disorders and quality of life outcomes in studies using primary care databases for understanding these disorders. The main finding of this study is the heterogeneity of definitions and lack of clarity around these, which means that studies of the same condition may not be comparable with each other. The paper is of high quality and very nearly ready for publication. I have just a few minor comments. 1) There is no reference to a study protocol - it is now usual for systematic reviews to start with an archived protocol to reduce the risk of bias introduced by making decisions as you go along. Was there a protocol in this case and could the authors put it in an online repository? 2) Because this is such a long and thorough piece of work it might be good to have a section either at the beginning or end of the discussion which highlights the main messages of the paper more clearly and couples these with some strong messages about future directions: heterogeneity and lack of information in studies means that they are not replicable or comparable which limits the way results can be pooled, or conclusions drawn from whole bodies of studies. Validation efforts are limited, so we do not know how well outcomes reflect true cases of the conditions under study. What should happen next? 3) Similarly, I think more could be made of the "Implications" section. As a community, how can we improve our work going forward to minimise these limitations? Authors being obliged to publish code lists would be helpful. Substantial investment in data quality evaluation in these databases seems warranted, but who would fund this? How can this be raised higher on funders' and researchers' agendas?
--

	4) Also in "Implications": Future work should look at GPs' conceptualisations of mental health problems as it is likely that they have a different, less categorical way of defining mental health concepts compared to specialists. Understanding their working "phenotypes" or "ontologies" would help to develop more meaningful case definition. 3) Also in "Implications": A big problem with validation in database research is having no way of quantifying the missed cases (i.e. sensitivity). How might having a more specific, but less sensitive case definition affect the outcomes of our research? Would the people who have e.g. depression who are detected by our case definition, differ systematically from those who are missed, thus biasing the results? There still seems to be no consensus on how to identify numbers of false negative cases, despite this problem being raised in the literature 8 years ago (see: Nicholson et al., What does validation of cases in electronic record databases mean? The potential contribution of free text. Pharmacoepidemiology and Drug Safety 2011; 20: 321–324). Some mention of this issue would be good, in the discussion, as it is likely to be a substantial problem in mental health research using primary care databases, due to the perceived stigma of the conditions. 4) Finally, would the authors consider putting all the code lists in their appendix in a repository such as "clinicalcodes.org" (with authors' permissions) in order to improve the clarity of case definitions going forward?
--	---

REVIEWER	Joyce So University of Toronto, Canada
REVIEW RETURNED	11-Feb-2019

GENERAL COMMENTS	This is a systematic review of mental health and quality of life outcomes in primary care databases in the UK. The authors conclude based on their study that there is heterogeneity and lack of validation in the use of codes to represent mental health and QoL outcomes in studies using EHRs from primary care databases in the UK. Overall, the study results could have wide-reaching implications for many studies that use EHR data. This is a systematic review of mental health and quality of life outcomes in primary care databases in the UK. The authors conclude based on their study that there is heterogeneity and lack of validation in the use of codes to represent mental health and QoL outcomes in studies using EHRs from primary care databases in the UK. Overall, the study results could have wide-reaching implications for many studies that use EHR data. Abstract: -"Objectives: To summarise the methods used to identify outcomes of anxiety, depression, fatigue, dementia/mild cognitive dysfunction, sexual dysfunction, pain, sleep disorders, and fatal and non-fatal self-harm in studies using electronic health records from primary care databases in the United Kingdom (UK)." - there does not seem to be any review of the methods, but rather the codes (or perhaps the category of codes) used by various studies to represent mental health and QOL outcomes. This should be clarified in the objectives, and strengths/limitations (e.g. use the wording from the Discussion – Results overview section "This review summarised the definitions and combinations of codes used to identify outcomes...".
---

	-Conclusions: the conclusions seem to re-hash the results. Perhaps they can re-word the conclusions to emphasize that their study results demonstrate a need for standardized definitions and validated codes for mental health and QOL outcomes. Introduction: -page 5, line 45/46 – correct “neurophatic” (I think this should read “neuropathic”) Methods: -for “mild cognitive impairment” as an outcome, how was “mild” defined for the purposes of identifying eligible studies? Why was the moderate to severe spectrum of cognitive impairment not included? -page 7, line 33/34 - “...any description related to the handling of past or prevalent episodes of these outcomes” (is prevalent the correct word here?) Results: -the text duplicates what is already presented in the tables. It would be much easier/cleaner to simply provide a description of what each table shows and then reserve the text in the Results section to describe primarily information that is not present in the tables and perhaps a broad summary of the results (e.g. some/all of the information currently included in the Discussion – Results overview section). Discussion: -currently, Results overview section does not provide a higher-level interpretation of the results for further discussion (e.g. “In this systematic review of 120 studies, the definitions and combinations of codes used to identify mental health and QOL outcomes from UK primary care databases was....”) -page 16, line 50/51 – “NICE issue guidelines...” should read “issued” -page 16, line 53/54 – ref 36 is inserted mid-sentence -page 17, line 10/11 – “...indications for which antidepressants were described” should presumably read “prescribed” -page 18, other outcomes – “10% and 20% of all patients with mild cognitive impairment are expected to convert to dementia”is this meant to read “between 10-20% of all patients....”? Otherwise, please clarify what the two % refer to. This seems to be a low turnover to dementia to justify using cognitive impairment and dementia as a composite outcome. Do the results change if mild cognitive impairment is NOT used to proxy for dementia? -page 18, other outcomes – re: chronic fatigue, “.....decreased during 2001 and 2013.....”, is this meant to read “between 2001 and 2013”? Otherwise, what would be the reason for a decrease in only those 2 years? -page 19, implications – “it is reasonable that this pattern applies to other conditions”is this implying that it is reasonable for this to occur or do you mean “it is reasonable to extrapolate that....”? It would seem preferable that this wide margin of variability not be the case when identifying a specific diagnosis. -conclusions: I think the conclusions need to be more clear that the study looked at a certain subset of mental health and QoL outcomes, and then suggest that results could/likely pertain more widely to all mental health/QoL outcomes. There should also be a clearer concluding remark about the implications of this research
--	---

	(e.g. in terms of ability to interpret study results based on the use of such coded data, comparability between studies, etc.).
--	---

VERSION 1 – AUTHOR RESPONSE

Reviewer #1

General comment:

'I congratulate the authors on conducting a timely and thorough piece of work on an important topic. This review of the definition of mental health outcomes in primary care databases will be a useful resource for many researchers in the UK and beyond.

The authors have tackled a huge topic in a systematic way, giving insight into the heterogeneity of definitions of mental health disorders and quality of life outcomes in studies using primary care databases for understanding these disorders.

The main finding of this study is the heterogeneity of definitions and lack of clarity around these, which means that studies of the same condition may not be comparable with each other. The paper is of high quality and very nearly ready for publication. I have just a few minor comments.'

Answer to comment:

Thank you very much for reviewing our paper and for your constructive feedback that certainly helped to improve our manuscript. Please see our answers below to each specific comment.

Specific comment #1:

'1) There is no reference to a study protocol - it is now usual for systematic reviews to start with an archived protocol to reduce the risk of bias introduced by making decisions as you go along. Was there a protocol in this case and could the authors put it in an online repository?'

Answer to comment #1:

A protocol was produced before the start of the systematic review. However we did not formally register this, and PROSPERO does not accept the registration of systematic reviews that have been completed, and thus we cannot do this retrospectively. We have now included the original protocol in appendix 1.

Specific comment #2:

'2) Because this is such a long and thorough piece of work it might be good to have a section either at the beginning or end of the discussion which highlights the main messages of the paper more clearly and couples these with some strong messages about future directions: heterogeneity and lack of information in studies means that they are not replicable or comparable which limits the way results can be pooled, or conclusions drawn from whole bodies of studies. Validation efforts are limited, so we do not know how well outcomes reflect true cases of the conditions under study. What should happen next?'

Answer to comment #2:

We agree, and have added to the first paragraph of the discussion (p15) to address these points (see underlined text below):

'This review summarised the definitions and combinations of codes used to identify outcomes of anxiety, depression, dementia and cognitive impairment, fatigue, pain, sexual dysfunction, sleep disorder and self-injurious behaviour in primary care databases of patients in the UK. The list of codes used in the original studies was obtained for approximately half of the papers. The lack of detailed information on the definition of the outcomes in most studies raises important questions as to whether studies can be replicated by others. In the studies where the code list was available, for all outcomes, there was substantial heterogeneity in the type of codes included (e.g. diagnoses and symptoms) and drugs selected to identify outcomes; for the remaining studies, the details provided in the original

publications suggest a similar pattern. We also noted considerable variability in the clinical definition of some outcomes (e.g. inclusion/exclusion of bipolar disorders in studies of depression). Validation of methods used to identify these outcomes was rarely carried out; where done, positive predictive values of case definitions were variable but mostly above 80%. To overcome these issues in the current context of limited number of studies with validation efforts, it is imperative that researchers develop, validate and make publicly available code lists for these outcomes.'

Specific comment #3:

'3) Similarly, I think more could be made of the "Implications" section. As a community, how can we improve our work going forward to minimise these limitations? Authors being obliged to publish code lists would be helpful. Substantial investment in data quality evaluation in these databases seems warranted, but who would fund this? How can this be raised higher on funders' and researchers' agendas?'

Answer to comment #3:

We agree that data quality assessments are needed for these databases. We have added the following paragraph to the implications section (pages 20 and 21):

'Primary care databases of electronic health records have made important contributions to medicine worldwide, particularly in the fields of infectious, respiratory and cardiovascular diseases. The burden of mental disorders in high-income countries has increased substantially in the last decades, and more research is needed to be better understand these conditions. Primary care databases of EHRs have potential to make huge contributions to this area but, for this to happen, we need coordinated efforts across funding and research organisations to improve data quality. For example, if scientific journals make a requirement of having publically available lists of codes, this would likely encourage researchers to spend more time defining the outcomes and potentially seek funding for validation studies, which in turn could increase the awareness of funding institutions for the importance of assessing data quality in projects using these data. In the meantime, transparency in the list of codes used to define these outcomes and reporting of sensitivity analysis with different lists of codes are key.'

Specific comment #4:

'4) Also in "Implications": Future work should look at GPs' conceptualisations of mental health problems as it is likely that they have a different, less categorical way of defining mental health concepts compared to specialists. Understanding their working "phenotypes" or "ontologies" would help to develop more meaningful case definition.'

Answer to comment #4:

We agree. The following sentences have been added to the implications section (pages 19 and 20):

'In addition, there is stigma associated with mental health disorders, which may result in under recording of some conditions in initial phases of the disease (BMJ Open 2016;6:e010746).'

'Future works also need to look at how GPs conceptualise mental health problems, as these are expected to have less stringent definitions than psychiatrists, and this could provide insights into more meaningful case definitions.'

Specific comment #5:

3) Also in "Implications": A big problem with validation in database research is having no way of quantifying the missed cases (i.e. sensitivity). How might having a more specific, but less sensitive case definition affect the outcomes of our research? Would the people who have e.g. depression who are detected by our case definition, differ systematically from those who are missed, thus biasing the results? There still seems to be no consensus on how to identify numbers of false negative cases, despite this problem being raised in the literature 8 years ago (see: Nicholson et al., What does

validation of cases in electronic record databases mean? The potential contribution of free text. *Pharmacoepidemiology and Drug Safety* 2011; 20: 321–324). Some mention of this issue would be good, in the discussion, as it is likely to be a substantial problem in mental health research using primary care databases, due to the perceived stigma of the conditions.

Answer to comment #5:

Thank you for this comment. We agree. This now being addressed in page 20:

A particular challenge with validation of primary care-based mental health outcomes is quantifying false negatives, which requires linkage to a high quality external source of information, to identify cases that may have been “missed” in primary care records. The Mental Health Dataset (MHDS), which includes individual patient records of adults seeking mental health services in secondary care and has recently been made available for linkage with CPRD primary care databases, represents an opportunity to assess the proportion of false negatives identified with the code lists, at least for more severe outcomes. Until then, sensitivity analysis using different lists of codes should be done, so that results can be compared and the impact of using different code lists evaluated. The consequences of underascertaining mental health outcomes are likely to depend on study design; in a cohort design this will not generally result in biased relative risks, whereas in a case-control context, a bias towards the null is likely.’

Specific comment #6:

‘4) Finally, would the authors consider putting all the code lists in their appendix in a repository such as "clinicalcodes.org" (with authors' permissions) in order to improve the clarity of case definitions going forward?’

Answer to comment #6:

Thank you for this suggestion. Unfortunately we do not have the resources to ask permission from all authors and upload the codes with respective metadata about the study. This would be an entire undertaking of its own and we feel that it is outside the scope of this study. We do provide the entire list of codes from all studies in the appendix of our paper, which will be available to all via open access. In addition, our experience with repositories of codes such as ‘clinicalcodes.org’ indicate that these are also affected by lack of resources, and not necessarily permanently supported (indeed our understanding is that ‘clinicalcodes.org’ itself is no longer supported for new uploads). We have clarified this point in the implications section of the discussion:

‘Repositories of lists of codes allow researchers to access codelists easily. However, these repositories also need funding to be maintained, which limits their stability and consequently their use.’

Reviewer #2

General comment:

‘This is a systematic review of mental health and quality of life outcomes in primary care databases in the UK. The authors conclude based on their study that there is heterogeneity and lack of validation in the use of codes to represent mental health and QoL outcomes in studies using EHRs from primary care databases in the UK. Overall, the study results could have wide-reaching implications for many studies that use EHR data.’

Answer to general comment:

Thank you very much for reviewing our paper and for your constructive feedback. Please see our answers below to each specific comment.

Specific comment #1:

Abstract:

-“Objectives: To summarise the methods used to identify outcomes of anxiety, depression, fatigue, dementia/mild cognitive dysfunction, sexual dysfunction, pain, sleep disorders, and fatal and non-fatal self-harm in studies using electronic health records from primary care databases in the United Kingdom (UK).” - there does not seem to be any review of the methods, but rather the codes (or perhaps the category of codes) used by various studies to represent mental health and QOL outcomes. This should be clarified in the objectives, and strengths/limitations (e.g. use the wording from the Discussion – Results overview section “This review summarised the definitions and combinations of codes used to identify outcomes...”).

Answer to comment #1:

We agree and changed the sentence as per suggestion of the reviewer. Please see abstract on page 2.

Specific comment #2:

-Conclusions: the conclusions seem to re-hash the results. Perhaps they can re-word the conclusions to emphasize that their study results demonstrate a need for standardized definitions and validated codes for mental health and QOL outcomes.

Answer to comment #2:

We agree. The new conclusion now reads:

‘Conclusions: There is a need for standardized definitions and validated list of codes to assess mental health and quality of life outcomes in primary care databases in the UK.’

Specific comment #3:

Introduction:

-page 5, line 45/46 – correct “neurophatic” (I think this should read “neuropathic”)

Answer to comment #3:

Done. Thank you for spotting this mistake.

Specific comment #4:

Methods:

-for “mild cognitive impairment” as an outcome, how was “mild” defined for the purposes of identifying eligible studies? Why was the moderate to severe spectrum of cognitive impairment not included?

Answer to comment #4:

We understand this was not sufficiently clear in the manuscript. We intended to characterise cognitive dysfunction in the spectrum of severities ranging from mild to grave; dementia by definition fits the latter category. For the purpose of identifying eligible studies, we searched the databases for studies looking at cognitive dysfunction regardless of the severity. For the purpose of selecting the eligible studies, we selected the studies that explicitly referred to mild cognitive dysfunction or dementia. We have clarified this throughout the manuscript.

Specific comment #5:

-page 7, line 33/34 - “...any description related to the handling of past or prevalent episodes of these outcomes” (is prevalent the correct word here?)

Answer to comment #5:

Yes. The word prevalent is being use to refer to participants who had the outcome at the date defined as the start of the observation period in longitudinal studies. We intended to summarise this information because mental health disorders such as depression have long treatment periods and it may not be always clear to ascertain if the patient has the outcome at the beginning of the study. This was clarified in the manuscript; please see page 7.

Specific comment #6:

Results:

-the text duplicates what is already presented in the tables. It would be much easier/cleaner to simply provide a description of what each table shows and then reserve the text in the Results section to describe primarily information that is not present in the tables and perhaps a broad summary of the results (e.g. some/all of the information currently included in the Discussion – Results overview section).

Answer to comment #6:

We described the results in a consistent way for all outcomes, highlighting the main messages from the tables. We feel that this structured approach to the reporting of the results adds value for drawing out key patterns from our extensive tables in a way that is easy to follow. However, we would be happy to work with the editors at the point of production if they feel this must be shortened.

Specific comment #7:

Discussion:

-currently, Results overview section does not provide a higher-level interpretation of the results for further discussion (e.g. “In this systematic review of 120 studies, the definitions and combinations of codes used to identify mental health and QoL outcomes from UK primary care databases was....”)

Answer to comment #7:

Thank you for this comment. We have added a few lines with a higher level overview of the results.

Please see page 9; the added sentences are also presented below:

‘A total of 120 studies were eligible for the systematic review; list of codes were obtained for nearly half of the studies. The definitions and combinations of codes used to identify mental health and QoL outcomes from UK primary care databases were heterogeneous for all outcomes; there was particular variability in the inclusion/exclusion of codes for symptoms of the mental disorders. Prescriptions were not frequently used as proxy for mental disorders. Validation efforts were rarely employed. Detailed results for each outcome are provided below.’

Specific comment #8:

-page 16, line 50/51 – “NICE issue guidelines...” should read “issued”

Answer to comment #8:

Done. Thank you for spotting this.

Specific comment #9:

-page 16, line 53/54 – ref 36 is inserted mid-sentence

Answer to comment #9:

We moved the reference to the end of the sentence. Thank you.

Specific comment #10:

-page 17, line 10/11 – “...indications for which antidepressants were described” should presumably read “prescribed”

Answer to comment #10:

Corrected, thank you.

Specific comment #11:

-page 18, other outcomes – “10% and 20% of all patients with mild cognitive impairment are expected to convert to dementia”is this meant to read “between 10-20% of all patients....”? Otherwise, please clarify what the two % refer to. This seems to be a low turnover to dementia to justify using

cognitive impairment and dementia as a composite outcome. Do the results change if mild cognitive impairment is NOT used to proxy for dementia?

Answer to comment #11:

We are sorry that the wording of the sentence wasn't clear. The sentence now reads 'between 10% and 20% of the patients with mild cognitive impairment'. We would like to clarify that we did not intend to use mild cognitive impairment a proxy for dementia. We intended to characterise cognitive dysfunction in the spectrum of severities ranging from mild to grave; dementia by definition fits the latter category. Please also refer to answer to our answer to comment #4.

Specific comment #12:

-page 18, other outcomes – re: chronic fatigue, “.....decreased during 2001 and 2013.....”, is this meant to read “between 2001 and 2013”? Otherwise, what would be the reason for a decrease in only those 2 years?

Answer to comment #12:

Yes, it should have read 'between 2001 and 2013'. This has been corrected. Please see page 18.

Specific comment #13:

-page 19, implications – “it is reasonable that this pattern applies to other conditions”is this implying that it is reasonable for this to occur or do you mean “it is reasonable to extrapolate that.....”? It would seem preferable that this wide margin of variability not be the case when identifying a specific diagnosis.

Answer to comment #13:

We are sorry this wasn't clear. We decided to remove this sentence from the manuscript to avoid confusion.

Specific comment #14:

-conclusions: I think the conclusions need to be more clear that the study looked at a certain subset of mental health and QoL outcomes, and then suggest that results could/likely pertain more widely to all mental health/QoL outcomes. There should also be a clearer concluding remark about the implications of this research (e.g. in terms of ability to interpret study results based on the use of such coded data, comparability between studies, etc.).

Answer to comment #14:

We agree. Please see changes to the conclusion on page 22 and below. We also changed the implications of this study, following reviewer #1 comments.

'Detailed information about codes used to identify outcomes of anxiety, depression, fatigue, dementia/mild cognitive dysfunction, sexual dysfunction, pain, sleep disorders, and fatal and non-fatal self-harm in studies using electronic health records from primary care databases in the United Kingdom (UK) was unavailable for around half of studies of these outcomes. Where available, there was substantial heterogeneity in the list of codes used to ascertain cases. Most studies did not validate case definitions, though when this was done, positive predictive values were generally high. This review focused on common mental health disorders and quality of life outcomes, but our conclusions are likely to be generalizable to other mental health outcomes. Caution is needed when interpreting and comparing results between studies, as heterogeneity in case definitions may be large. Future studies should fully report outcomes definitions, use sensitivity analysis to mitigate uncertainties about the impact of the case definition on studies' reported outcomes, and seek to validate the list of codes used to identify these outcomes.'

VERSION 2 – REVIEW

REVIEWER	Elizabeth Ford Brighton and Sussex Medical School, United Kingdom
REVIEW RETURNED	27-Mar-2019
GENERAL COMMENTS	I am content that authors have addressed my comments and have nothing further to add.